# AUBER: AUTOMATED BERT REGULARIZATION

## ABSTRACT

How can we effectively regularize BERT? Although BERT proves its effectiveness in various downstream natural language processing tasks, it often overfits when there are only a small number of training instances. A promising direction to regularize BERT is based on pruning its attention heads based on a proxy score for head importance. However, heuristic-based methods are usually suboptimal since they predetermine the order by which attention heads are pruned. In order to overcome such a limitation, we propose AUBER, an effective regularization method that leverages reinforcement learning to automatically prune attention heads from BERT. Instead of depending on heuristics or rule-based policies, AUBER learns a pruning policy that determines which attention heads should or should not be pruned for regularization. Experimental results show that AUBER outperforms existing pruning methods by achieving up to $9.39\%$ better accuracy. In addition, our ablation study empirically demonstrates the effectiveness of our design choices for AUBER.

## 1 INTRODUCTION

How can we effectively regularize BERT (Devlin et al. (2018))? In natural language processing (NLP), fine-tuning a large-scale pre-trained language model has greatly enhanced generalization. In particular, BERT has demonstrated effectiveness through improvements in many downstream NLP tasks such as sentence classification and question answering.

Despite its recent success and wide adoption, fine-tuning BERT on a downstream task is prone to overfitting due to over-parameterization; BERT-base has 110M parameters and BERT-large has 340M parameters. The overfitting worsens when the target downstream task only has a small number of training examples. Devlin et al. (2018) and Phang et al. (2018) show that datasets with 10,000 or less training examples sometimes fail to fine-tune BERT.

To mitigate this critical issue, multiple studies attempt to regularize BERT by pruning parameters or using dropout to decrease its model complexity (Michel et al. (2019); Voita et al. (2019); Lee et al. (2020)). Among these approaches, we regularize BERT by pruning attention heads since pruning yields simple and explainable results and it can be used along with other regularization methods. In order to avoid combinatorial search, whose computational complexity grows exponentially with the number of heads, the existing methods measure the importance of each attention head based on heuristics such as an approximation of sensitivity of BERT to pruning a specific attention head. However, these approaches are based on hand-crafted heuristics that are not guaranteed to be directly related to the model performance, and therefore, would result in a suboptimal performance.

In this paper, we propose AUBER, an effective method for regularizing BERT. AUBER overcomes the limitation of past attempts to prune attention heads from BERT by leveraging reinforcement learning. When pruning attention heads from BERT, our method automates this process by learning policies rather than relying on a predetermined rule-based policy and heuristics. AUBER prunes BERT sequentially in a layer-wise manner. For each layer, AUBER extracts features that are useful for the reinforcement learning agent to determine which attention head to be pruned from the current layer. The final pruning policy found by the reinforcement learning agent is used to prune the corresponding layer. Before AUBER proceeds to process the next layer, BERT is fine-tuned to recapture the information lost due to pruning attention heads. An overview of AUBER transitioning from the second to the third layer of BERT is demonstrated in Figure 1.

Our contributions are summarized as follows:

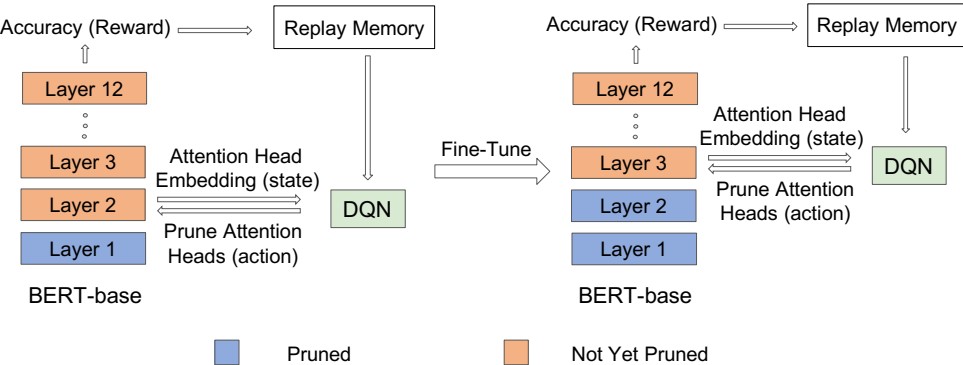

Figure 1: An overview of AUBER transitioning from $Layer$ 2 to $Layer$ 3 of BERT-base.

- **Method.** We propose AUBER for automatically learning to regularize BERT by reinforcement learning. AUBER is designed to carefully represent the state of BERT and reduce the action search cost by dually-greedy search space pruning.
- **Analysis.** We theoretically justify our design choice of using $L1$ norm of the value matrix of each attention head as an element of a state (see Theorem 1).
- **Experiments.** We perform extensive experiments, and show that AUBER successfully regularizes BERT improving the performance by up to $9.39\%$ and outperforms other head pruning methods. Through ablation study, we empirically show that our design choices for AUBER are effective.

## 2 PRELIMINARY

We describe preliminaries on multi-headed self-attention (Section 2.1), BERT (Section 2.2), and deep Q-learning (Section 2.3).

### 2.1 MULTI-HEADED SELF-ATTENTION

An attention function maps a query vector and a set of key-value vector pairs to an output. We compute the query, key, and value vectors by multiplying the input embeddings $Q, K, V \in \mathbb{R}^{N \times d}$ with the parameterized matrices $W^Q \in \mathbb{R}^{d \times n}$, $W^K \in \mathbb{R}^{d \times n}$, and $W^V \in \mathbb{R}^{d \times m}$ respectively, where $N$ is the number of tokens in the sentence, and $n, m$, and $d$ are query, value, and embedding dimension respectively. In multi-headed attention, $H$ independently parameterized attention heads are applied in parallel to project the input embeddings into multiple representation subspaces. Each attention head contains parameter matrices $W_i^Q \in \mathbb{R}^{d \times n}$, $W_i^K \in \mathbb{R}^{d \times n}$, and $W_i^V \in \mathbb{R}^{d \times m}$. Output matrices of $H$ independent attention heads are concatenated and projected by a matrix $W^O \in \mathbb{R}^{Hm \times d}$ to obtain the final result. This process can be represented as:

$$MultiHeadAtt(Q, K, V) = Concat(Att_{1...H}(Q, K, V))W^O, \tag{1}$$

where

$$Att_i(Q, K, V) = softmax(\frac{(QW_i^Q)(KW_i^K)^T}{\sqrt{n}})VW_i^V. \tag{2}$$

A multi-headed self-attention follows the same mapping methods as general multi-headed attention function except that all the query, key, and value embeddings come from the same sequence.

### 2.2 BERT

BERT (Devlin et al. (2018)) is a multi-layer Transformer (Vaswani et al. (2017)) pre-trained on masked language model and next sentence prediction tasks. It is then fine-tuned on specific tasks including language inference and question answering. BERT-base has 12 Transformer layers and each layer has 12 self-attention heads. Despite its success in various NLP tasks, BERT sometimes overfits when the training dataset is small due to over-parameterization. Thus, there has been a growing interest in BERT regularization through various methods such as dropout (Lee et al. (2020)).

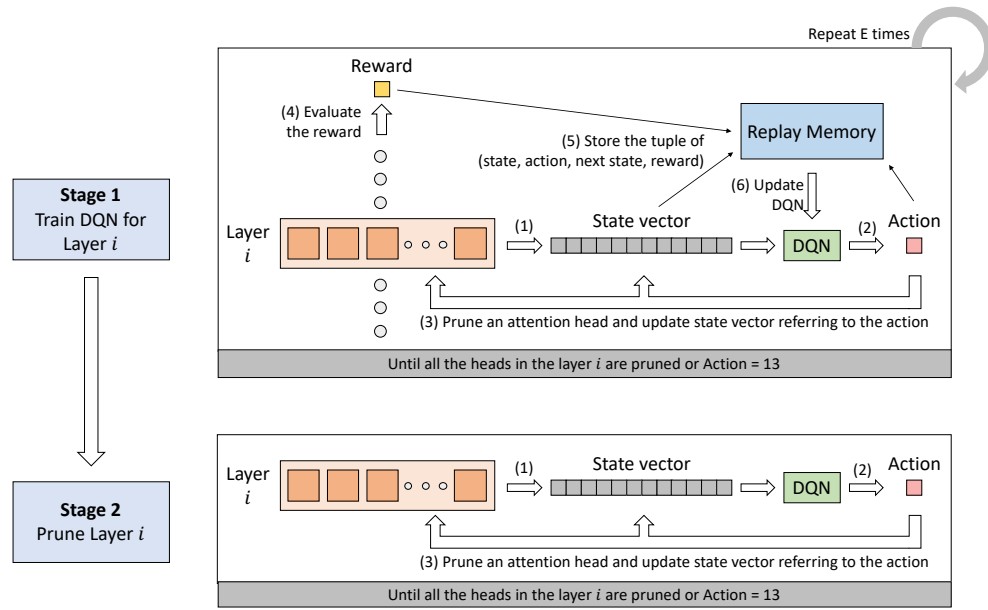

Figure 2: Overall flow of AUBER while pruning layer $i$. It first trains DQN and then uses the trained DQN to make the final decision on which heads to prune.

## 2.3 DEEP Q-LEARNING

A deep Q network (DQN) is a multi-layer neural network that outputs a vector of action-value pairs for a given state $s$. It is a function that maps a $d_s$-dimensional state space to a $d_a$-dimensional action space. Two important features of the DQN algorithm are target network and experience replay (Mnih et al. (2013)). The target network has the same architecture as that of the policy network, and its parameters are copied every $\tau$ steps from the policy network. For experience replay, transition tuples (i.e. (state, action, reward, next state)) are stored in a first-in-first-out memory buffer, and a subset of those tuples, uniformly sampled from the buffer, update the parameters of the policy network.

## 3 PROPOSED METHOD

We propose AUBER, our method for regularizing BERT by automatically learning to prune attention heads from BERT. After presenting the overview of the proposed method in Section 3.1, we describe how we frame the problem of pruning attention heads into a reinforcement learning problem in Section 3.2. Section 3.3 explains how states are represented in AUBER and provides a justification for the process. Section 3.4 describes how AUBER reduces the extremely large search space, and describe the full algorithm.

## 3.1 OVERVIEW

We observe that BERT is prone to overfitting for tasks with a few training data. However, the existing head pruning methods rely on hand-crafted heuristics and hyperparameters, which give sub-optimal results. The goal of AUBER is to automate the pruning process for successful regularization. Designing such regularization method entails the following challenges:

1. **Automation.** How can we automate the head pruning process for regularization without resorting to sub-optimal heuristics?
2. **State representation.** When automating the regularization process as a reinforcement learning problem, how can we represent the state of BERT in a way useful for the pruning?
3. **Action search space scalability.** BERT has many parameters and attention heads in each layer. How can we handle prohibitively large action search space for pruning?

We propose the following main ideas to address the challenges:

1. **Regularization with reinforcement learning (Section 3.2).** We exploit reinforcement learning, specifically DQN, with accuracy enhancement as reward. DQN has shown a superior performance for many tasks, and is a natural choice for model-free and off-policy learning (Sutton & Barto (2018)), which is exactly our setting. Experience replay also allows efficient usage of previous experiences and stable convergence (Mnih et al. (2013)).
2. **L1 norm of value matrix as state representation (Section 3.3).** We use L1 norm of value matrix of each attention head to represent the state of a layer. When a head is pruned, the corresponding value is set to 0.
3. **Dually-greedy search space pruning (Section 3.4).** To reduce the search space, we use two greedy methods: 1) we prune layer-by-layer, and an action is performed only within a layer, and 2) in each layer, we prune one attention head at a time to reduce the search space.

## 3.2 AUTOMATED REGULARIZATION WITH REINFORCEMENT LEARNING

AUBER leverages reinforcement learning for efficient search of regularization strategy without relying on heuristics. We exploit DQN among various reinforcement learning frameworks which have shown a superior performance in model-free and off-policy environment. The overall flow is described in Figure 2. In this section, we introduce the detailed setting of our reinforcement learning framework.

**Initial state.** Each layer of BERT has multiple attention heads, each of which has its own query, key, and value matrices. For layer $l$ of BERT, we derive the initial state $s_l$ using L1 norm of the value matrix of each attention head. Further details for this computation is elaborated in Section 3.3.

**Action.** The action space $a$ of AUBER is discrete. For a BERT model with $H$ attention heads per layer, the number of possible actions is $H + 1$ (i.e. $a \in \{1, 2, \ldots, H, H + 1\}$). When the action $a = i \in \{1, 2, \ldots, H - 1, H\}$ is chosen, the corresponding $i^{th}$ attention head is pruned. The action $a = H + 1$ signals the DQN agent to quit pruning. To facilitate exploration via off-policy learning, actions are chosen in a decaying-epsilon-greedy manner: the agent chooses a random action with probability $\epsilon$ that decays over time or otherwise selects an action based on the current policy, where

$$\epsilon = \epsilon_{final} + \frac{\epsilon_{initial} - \epsilon_{final}}{e^{s/\epsilon_{decay}}} \tag{3}$$

Here, $s$ is the total number of actions taken by the agent up to the current episode, $\epsilon_{initial}$ is the starting value of $\epsilon$ (i.e. when $s = 0$), $\epsilon_{final}$ is the value that $\epsilon$ converges to as $s \to \infty$, and $\epsilon_{decay}$ is a hyperparameter that adjusts the rate of decay of $\epsilon$.

**Next State.** After the $i^{th}$ head is pruned, the value of $i^{th}$ index of $s_l$ is set to 0. This modified state is provided as the next state to the agent. This mechanism allows the agent to recognize which attention heads have been pruned and decide the next best pruning policy based on past decisions. When the action $a = H + 1$, the next state is set to the terminating state which ends an episode.

**Reward.** To evaluate the reward, the training data split into two sets: mini-training set and mini-dev set. We use the data in mini-training set for fine-tuning and data in mini-dev set for reward evaluation. The reward of AUBER is the change in accuracy

$$\Delta acc = current\_accuracy - prev\_accuracy, \tag{4}$$

where $current\_accuracy$ is the accuracy of the current BERT model evaluated on the mini-dev set, and $prev\_accuracy$ is the accuracy obtained from the previous state or the accuracy of the original BERT model if no attention heads are pruned.

If we set the reward simply as $current\_accuracy$, DQN cannot capture the differences among reward values if the changes in accuracy are relatively small. Setting the reward as the change in accuracy has the normalization effect, thus stabilizing the training process of the DQN agent. The reward for action $a = H + 1$ is a hyper-parameter that can be adjusted to encourage or discourage active pruning. In AUBER, it is set to $0$ to encourage the DQN agent to prune only when the expected change in accuracy is positive.

**Fine-tuning.** After the best pruning policy for layer $l$ of BERT is found, the BERT model pruned according to the best pruning policy is fine-tuned with a smaller learning rate. This fine-tuning step is crucial since it adjusts the weights of remaining attention heads to compensate for the information lost due to pruning. Then, the initial state of the layer $l + 1$ is calculated and provided to the agent. Since frequent fine-tuning may lead to overfitting, we separate the training dataset into two: a mini-

validation dataset and a mini-training dataset. The mini-validation dataset is the dataset on which the pruned BERT model is evaluated on to return a reward. After the optimal pruning policy is determined by using the mini-validation dataset, the mini-training dataset is used to fine-tune the pruned model. When all layers are pruned by AUBER, the final model is fine-tuned with the entire training dataset with early stopping.

### 3.3 STATE REPRESENTATION

The initial state $s_l$ of layer $l$ of BERT is computed through the following procedure. We first calculate the L1 norm of the value matrix of each attention head. Then, we standardize the norm values to have a mean $\mu = 0$ and a standard deviation $\sigma = 1$. Finally, the $softmax$ function is applied to the norm values to yield $s_l$. The justification of using $L1$ norm is given by Theorem 1 (proof in Appendix A.1) that L1 norm of the value matrix of a head bounds the L1 norm of its output matrix, which implies the importance of the head in the layer.

**Theorem 1.** *For a layer with $H$ heads, let $N$ be the number of tokens in the sentence and $m$, $n$, and $d$ be the value, query, and embedding dimension respectively. Let $Q, K, V \in \mathbb{R}^{N \times d}$ be the input query, key, and value matrices, and $W_i^Q$, $W_i^K$, and $W_i^V$ be the weight parameters of the $i^{th}$ head such that $W_i^Q, W_i^K \in \mathbb{R}^{d \times n}$ and $W_i^V \in \mathbb{R}^{d \times m}$. Let $O_i$ be the output of the $i^{th}$ head. Then, $\|O_i\|_1 \leq C\|W_i^V\|_1$ where the constant $C = N\|V\|_1$.* $\square$

---

**Algorithm 1:** AUBER: Automatic BERT Regularization

---

**Input** : A BERT model $B_t$ fine-tuned on task $t$, $\# L$ of layers in BERT model, $\# H$ of attention heads per layer of BERT model, and $\# E$ of episodes.
**Output:** Regularized $B_t$.

1 **for** $l \leftarrow 1$ **to** $L$ **do**
2      Initialize policy network $P$ and replay memory $M$
3      **for** $e \leftarrow 1$ **to** $E$ **do**
4          $B_t^* \leftarrow copy(B_t)$
5          $s_l \leftarrow B_t^*.state\_vector(l)$
6          $prev\_accuracy \leftarrow eval(B_t^*)$
7          **while** $action \neq H + 1$ **do**
8              **if** $B_t^*.one\_head\_left$ **then**
9                  $action \leftarrow H + 1$
10              **else**
11                  $action \leftarrow P.choose\_action(s_l)$
12              **end**
13              **if** $action = H + 1$ **then**
14                  $s_l^* \leftarrow Terminal\ State$
15                  $reward \leftarrow 0$
16              **else**
17                  $B_t^*.prune\_head(action)$
18                  $s_l^* \leftarrow copy(s_l)$
19                  $s_l^*[action] \leftarrow 0$
20                  $current\_accuracy \leftarrow eval(B_t^*)$
21                  $reward \leftarrow current\_accuracy - prev\_accuracy$
22                  $prev\_accuracy \leftarrow current\_accuracy$
23              **end**
24              $M.push(s_l, action, s_l^*, reward)$
25              $s_l \leftarrow s_l^*$
26          **end**
27          $P.optimize(M)$
28      **end**
29      $B_t.prune(P.final\_policy(l))$
30      $B_t.finetune()$
31 **end**

---

### 3.4 DUALLY-GREEDY SEARCH SPACE PRUNING

The total number of attention heads in 12-layer BERT is 144, when 12 attention heads are used. Naively designing actions would lead to $2^{144}$ possible actions which are prohibitively large. Our idea to reduce the search space is dually-greedy pruning: we prune layer-by-layer in a greedy manner (from lower to upper layers), and in each layer we greedily prune 1 attention head at a time.

For each layer $l$ with $H$ attention heads, the agent receives an initial layer embedding $s_l$ which encodes useful characteristics ($l_1$ norm of the attention heads) of this layer. Then, the agent outputs the index of an attention head that is expected to increase the training accuracy when removed. After an attention head $i$ is pruned, the value of the $i^{th}$ index of $s_l$ is set to 0, and it is provided as the next state to the agent. This process is repeated until the action $a$ becomes $H + 1$. The model pruned up to layer $l$ is fine-tuned on the training dataset, and a new initial layer embedding $s_{l+1}$ is calculated from the fine-tuned model.

Algorithm 1 illustrates the process of AUBER. Algorithm 1 receives a BERT model $B_t$, which is fine-tuned on a specific task $t$, and the parameters for BERT and reinforcement learning. The algorithm aims to output the regularized $B_t$. In line 2, we initialize a policy network $P$ and a replay memory $M$ for layer $l$. Lines 3-28 train the policy network $P$ for $E$ episodes. For each episode, lines 8-25 choose an $action$ based on the current state vector $s_l$, prune an attention head based on the $action$, compute the resulting $reward$ and the next state $s_l^*$, and store the transition tuple into $M$. This process is repeated until $action = H + 1$, which indicates the termination of pruning. In line 27, we optimize $P$ with transition tuples sampled from $M$. Finally, in lines 29-30, we use the trained policy network to find the optimal pruning policy for layer $l$, prune $B_t$ according to the policy, and finally fine-tune $B_t$. After pruning a layer $l$, we proceed to prune the next layer $l + 1$ up to layer $L$.

## 4 EXPERIMENTS

We conduct experiments to answer the following questions of AUBER.

**Q1 Accuracy (Section 4.2).** Given a BERT model fine-tuned on a specific NLP task, how well does AUBER improve the accuracy of the model?

**Q2 State Representation (Section 4.3).** How useful is the *L1 norm of the value matrices* of attention heads in representing the state of BERT?

**Q3 Order of Processing Layers (Section 4.4).** How does the order in which the layers are processed by AUBER affect regularization?

### 4.1 EXPERIMENTAL SETUP

**Datasets.** We test AUBER on four GLUE datasets (Wang et al. (2019)) - MRPC, CoLA, RTE, and WNLI which contain less than 10,000 training instances; it has been observed that datasets with 10,000 or less training examples often fail in fine-tuning BERT (Devlin et al. (2018); Phang et al. (2018)). Detailed information of these datasets is described in Table 1.

Table 1: Datasets.

| dataset | # of classes | # of train | # of dev |
|---------|--------------|------------|----------|
| MRPC[1] | 2 | 3668 | 408 |
| CoLA[2] | 2 | 8551 | 1043 |
| RTE[3]  | 2 | 2490 | 277 |
| WNLI[4] | 2 | 635 | 71 |

---

[1] https://www.microsoft.com/en-us/download/details.aspx?id=52398
[2] https://nyu-mll.github.io/CoLA/
[3] https://aclweb.org/aclwiki/Recognizing_Textual_Entailment
[4] https://cs.nyu.edu/faculty/davise/papers/WinogradSchemas/WS.html

Table 2: Performance of AUBER and competitors on 4 GLUE tasks. The performance after fine-tuning is measured. AUBER gives the best performance for the same number of pruned heads. Bold font indicates the best accuracy among competing pruning methods.

|                     | MRPC                  | CoLA                  | RTE                   | WNLI                  | Average               |
|---------------------|-----------------------|-----------------------|-----------------------|-----------------------|-----------------------|
| Original            | 84.07                 | 57.01                 | 63.54                 | 46.48                 | 62.77                 |
| AUBER               | **85.62±0.51**        | **60.59±0.73**        | **65.34±1.30**        | **56.06±0.63**        | **66.90±0.79**        |
| Random              | 84.02±1.12            | 57.89±0.90            | 63.47±1.29            | 54.08±2.14            | 64.86±1.36            |
| Confidence          | 83.70±0.47            | 57.69±2.19            | 64.26±1.64            | 55.77±0.77            | 65.36±1.27            |
| Michel et al. (2019)| 84.22±0.33            | 58.86±0.64            | 63.90±0.00            | 55.21±1.84            | 65.55±0.70            |
| Voita et al. (2019) | 83.92±0.71            | 55.34±0.81            | 64.12±1.65            | 52.96±5.51            | 64.08±2.17            |

**BERT Model.** We use the pre-trained *bert-base-cased* model with 12 layers and 12 attention heads per layer provided by huggingface[5]. We fine-tune this model on each dataset mentioned in Table 1 to obtain the initial model. Initial models for MRPC, CoLA, and WNLI are fine-tuned on the corresponding dataset for 3 epochs, and the initial model for RTE is fine-tuned for 4 epochs. The maximum sequence length is set to 128, and the batch size per GPU is set to 32. The learning rate for fine-tuning initial models for MRPC, CoLA, and WNLI is set to 0.00002, and the learning rate for fine-tuning the initial model for RTE is set to 0.00001.

**Reinforcement Learning.** We use a 4-layer feedforward neural network for the DQN agent. The dimensions of input, output, and all hidden layers are set to 12, 13, and 512, respectively. LeakyReLU is applied after all layers except for the last one. We train the DQN agent for 150 episodes. For the epsilon greedy strategy for choosing actions, $\epsilon_{initial}$ and $\epsilon_{final}$ are set to 1 and 0.05 respectively, and the epsilon decreases exponentially with the decay rate $\epsilon_{decay}$ of 256. The replay memory size is set to 5000, and the batch size for training the DQN agent is set to 128. The discount value $\gamma$ for the DQN agent is set to 1. The learning rate is set to 0.000002 when fine-tuning BERT after processing a layer. Before processing each layer, the training dataset is randomly split into 1 : 2 to yield a mini-training dataset and a mini-validation dataset. When fine-tuning the final model, the patience value of early stopping is set to 20.

**Competitors.** We compare AUBER with other methods that prune BERT's attention heads. If AUBER prunes $P$ number of attention heads from BERT, we prune $P$ heads in all the competitors.

- **Random.** Prune attention heads randomly.
- **Confidence.** Prune $P$ heads with the smallest confidence score, which is the average of the maximum attention weight after a series of forward passes. A high confidence score indicates that the weight is concentrated on a single token.
- **Michel et al. (2019).** Perform a forward and backward pass to calculate gradients and use them to assign an importance score to each attention head.
- **Voita et al. (2019).** Construct a new loss function that minimizes both the classification error and the number of used heads so that unproductive heads are pruned while maintaining the model performance.

We fine-tune the competitors after pruning in layer-wise manner for fair comparison. In other words, we prune the attention heads in layer-wise manner and fine-tune the model every layer.

**Implementation.** We construct all models using PyTorch framework. All the models are trained and tested on a GeForce GTX 1080 Ti GPU.

## 4.2 ACCURACY

We evaluate the performance of AUBER against competitors. Table 2 shows the results on four GLUE datasets listed in Table 1. Note that AUBER outperforms all of its competitors on regularizing BERT, providing the best accuracy for all the datasets. While most of the competitors fail to improve performance of BERT on the dev dataset of MRPC and CoLA, AUBER improves the performance of BERT by up to 9.39% on those datasets.

---

[5] https://github.com/huggingface/transformers

Table 3: We compare AUBER with four variants: AUBER-Query, AUBER-Key, AUBER-L2, and AUBER-Reverse on four GLUE datasets. AUBER-Query and AUBER-Key use the query and key matrices respectively, and AUBER-L2 uses the L2 norm of the value matrix to obtain the initial state. AUBER-Reverse processes BERT starting from the final layer (e.g. $12^{th}$ layer for BERT-base). Bold font indicates the best accuracy among competing pruning methods.

(a) MRPC (Accuracy)

| Policy | # of heads pruned | dev |
|---|---|---|
| AUBER | 20 | 86.03 |
| AUBER-Query | 27 | 83.33 |
| AUBER-Key | 42 | 84.56 |
| AUBER-L2 | 33 | 83.33 |
| AUBER-Reverse | 58 | **86.52** |

(b) CoLA (Matthew's correlation)

| Policy | # of heads pruned | dev |
|---|---|---|
| AUBER | 53 | **61.28** |
| AUBER-Query | 72 | 55.52 |
| AUBER-Key | 55 | 55.78 |
| AUBER-L2 | 63 | 54.85 |
| AUBER-Reverse | 57 | 59.48 |

(c) RTE (Accuracy)

| Policy | # of heads pruned | dev |
|---|---|---|
| AUBER | 87 | **66.43** |
| AUBER-Query | 83 | 65.34 |
| AUBER-Key | 86 | 63.18 |
| AUBER-L2 | 61 | **66.43** |
| AUBER-Reverse | 99 | 64.62 |

(d) WNLI (Accuracy)

| Policy | # of heads pruned | dev |
|---|---|---|
| AUBER | 86 | **56.34** |
| AUBER-Query | 96 | **56.34** |
| AUBER-Key | 101 | **56.34** |
| AUBER-L2 | 94 | 53.52 |
| AUBER-Reverse | 101 | 54.93 |

### 4.3 EFFECT OF STATE REPRESENTATION

We validate that the *L1 norm of value matrix* of each attention head effectively guides AUBER to predict the best action. Table 3 shows the performances of the variants of AUBER.

#### 4.3.1 AUBER WITH THE KEY/QUERY MATRICES AS THE STATE VECTOR

Among the query, key, and value matrices of each attention head, we show that the value matrix best represents the current state of BERT. Here we evaluate the performance of AUBER against AUBER-Query and AUBER-Key. AUBER-Query and AUBER-Key use the query and key matrices respectively to obtain the initial state. Note that AUBER, which uses the value matrix to obtain state vectors, outperforms AUBER-Query and AUBER-Key on all four tasks.

#### 4.3.2 AUBER WITH L2 NORM OF THE VALUE MATRICES AS THE STATE VECTOR

L1 norm of the value matrices is used to compute the state vector based on the theoretical derivation. In this ablation study, we experimentally show that the L1 norm of the value matrices is appropriate for state vector. We set a new variant AUBER-L2 which leverages L2 norm of the value matrices to compute the initial state vector instead of L1 norm. The performance of AUBER is far more superior than AUBER-L2 in most cases bolstering that L1 norm of the value matrices effectively represents the state of BERT.

### 4.4 EFFECT OF ORDER OF PROCESSING LAYERS

We empirically demonstrate how the order in which the layers are processed affects the final performance. We evaluate the performance of AUBER against AUBER-Reverse which processes BERT layers in the opposite direction (i.e. starting from the $12^{th}$ layer) to what AUBER does. As shown in Table 3, AUBER provides a better performance than AUBER-Reverse in 3 out of 4 cases. This shows that pruning lower layers first and then move to upper layers by AUBER is effective. A possible explanation is that it is easier to train task-specific parameters (those in upper layers) after learning general parameters (those in lower layers), rather than that in the reverse order.

## 5 RELATED WORK

**BERT Regularization.** To prevent overfitting of BERT on downstream NLP tasks, various regularization techniques have been proposed. Variants of dropout improves the stability of fine-tuning large pre-trained language models even when presented with a small number of training examples (Lee et al. (2020); Fan et al. (2020)). Using slanted triangular learning rate schedule and discriminative fine-tuning has been proven to effectively prevent overfitting (Howard & Ruder (2018)). Jiang et al. (2019) propose SMART, which regularizes the model by smoothing it and preventing aggressive update while Liu et al. (2019) train BERT model with multi-task learning algorithm to remedy overfitting. Introducing adversarial training to enhance the generalization of BERT has been tackled in Zhu et al. (2020). Pre-training tasks have also been changed for a better regularization. Xu et al. (2020) modify the pre-training task, next sentence prediction, to additionally predict previous sentence so that it can capture more correlation. Our method has definite advantages for it can be used along with any of the aforementioned methods.

**BERT Pruning.** A number of studies have analyzed the effectiveness of pruning parameters in BERT. Gordon et al. (2020) experimentally prove overparameterization of BERT by showing that pruning $30 - 40\%$ of parameters does not affect model performance. Chen et al. (2020) find out the best parameter pruning strategy from the viewpoint of the lottery ticket hypothesis while Guo et al. (2019) deploy reweighted L1 regularization with proximal algorithm. However, these methods primarily aim to compress BERT model not to regularize it, and, therefore, no specific method to enhance the model performance has been proposed.

Thanks to the unique structure of BERT that consists of multi-headed attention, not only parameter pruning but also structured attention head pruning has been addressed (Voita et al. (2019); Michel et al. (2019); Kovaleva et al.; McCarley (2019)). Michel et al. (2019); Kovaleva et al.; Prasanna et al. (2020) evaluate the role and importance of each attention head by measuring heuristics: the average of its maximum attention weight, where average is taken over tokens in a set of sentences used for evaluation, or the expected sensitivity of the model to attention head pruning. Their results show that a large percentage of attention heads with low importance scores can be pruned without significantly impacting performance. The approach to set L0 regularization term to minimize both the training loss and the number of used attention heads has been also presented (Voita et al. (2019); McCarley (2019)). However, they usually yield suboptimal results since they predetermine the order in which the attention heads are pruned by using heuristics.

**Automation of Neural Network Pruning** To automate the process of Convolutional Neural Network pruning, Lin et al.; He & Han (2018) have leveraged reinforcement learning to determine the best pruning strategy for each layer. Important features that characterize a layer are provided to a reinforcement learning agent to determine how much of the current layer should be pruned. To the best of our knowledge, AUBER is the first attempt to use reinforcement learning to prune attention heads from Transformer-based models such as BERT.

## 6 CONCLUSION

We propose AUBER, an effective method to regularize BERT by automatically pruning attention heads. Instead of depending on heuristics or rule-based policies, AUBER leverages reinforcement learning to learn a pruning policy that determines which attention heads should be pruned for better regularization. Experimental results demonstrate that AUBER effectively regularizes BERT, increasing the performance of the original model on the dev dataset by up to $9.39\%$. In addition, we experimentally demonstrate the effectiveness of our design choices for AUBER.

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

## A APPENDIX

### A.1 PROOF FOR THEOREM 1

*Proof.* For $i^{th}$ head in the layer, let

$$softmax_i = softmax(\frac{(QW_i^Q)(KW_i^K)^T}{\sqrt{n}}) \tag{5}$$

and

$$v_i = VW_i^V. \tag{6}$$

The output of the head, $O_i$, is evaluated as $O_i = softmax_i v_i$. Then,

$$\|O_i\|_1 = \sum_{j=1}^{N}\sum_{k=1}^{m}|(O_i)_{jk}| \tag{7}$$

$$= \sum_{j=1}^{N}\sum_{k=1}^{m}|((softmax_i)_{j\cdot})^T(v_i)_{\cdot k}| \tag{8}$$

$$\leq \sum_{j=1}^{N}\sum_{k=1}^{m}\|(softmax_i)_{j\cdot}\|_2\|(v_i)_{\cdot k}\|_2 \tag{9}$$

$$= \sum_{j=1}^{N}\|(softmax_i)_{j\cdot}\|_2\sum_{k=1}^{m}\|(v_i)_{\cdot k}\|_2 \tag{10}$$

Since the L1 norm of a vector is always greater than or equal to the L2 norm of the vector,

$$\|O_i\|_1 \leq \sum_{j=1}^{N}\|(softmax_i)_{j\cdot}\|_1\sum_{k=1}^{m}\|(v_i)_{\cdot k}\|_1 \tag{11}$$

$$= N\sum_{k=1}^{m}\|(v_i)_{\cdot k}\|_1 \tag{12}$$

$$= N\sum_{j=1}^{N}\sum_{k=1}^{m}|(v_i)_{jk}| \tag{13}$$

$$= N\sum_{j=1}^{N}\sum_{k=1}^{m}|(V_{j\cdot})^T(W_i^V)_{\cdot k}| \tag{14}$$

$$\leq N\sum_{j=1}^{N}\sum_{k=1}^{m}\|V_{j\cdot}\|_2\|(W_i^V)_{\cdot k}\|_2 \tag{15}$$

$$= N\sum_{j=1}^{N}\|V_{j\cdot}\|_2\sum_{k=1}^{m}\|(W_i^V)_{\cdot k}\|_2 \tag{16}$$

$$\leq N\sum_{j=1}^{N}\|V_{j\cdot}\|_1\sum_{k=1}^{m}\|(W_i^V)_{\cdot k}\|_1 \tag{17}$$

$$= N\|V\|_1\|W_i^V\|_1. \tag{18}$$

where the norm of the matrices is entrywise norm, $\|A\|_1 = \sum_j\sum_k A_{jk}$. All heads in the same layer take the same $V$ as input and $N$ is constant. Thus,

$$\|O^i\|_1 \leq C\|W_i^V\|_1 \tag{19}$$

for the constant $C = N\|V\|_1$. □

