# OpenReview forum: "AUBER: Automated BERT Regularization"
_ICLR.cc/2021/Conference — Reject_

### Official Review · AnonReviewer2 · 2020-10-28

**Rating:** 5
**Confidence:** 5

**Review:**

Summary:

The paper focuses on reducing over-fitting for the BERT model by pruning the attention heads. For determining the order of pruning, the paper uses reinforcement learning along with greedy algorithm to get rid of unimportant attention heads. The experimental section shows the analysis on various design choices like using value matrix to represent the state instead of key or query matrices, using L1 norm of value matrix rather than L2 norm and lastly, the order in which the layers should  be handled I.e. from top to  bottom or the reverse.

+ve

- The paper is easy to follow and the authors have presented the details in smaller sub-sections which makes it easy to understand.
- Using RL to determine the head pruning sequence makes if better than the previous greedy methods like Michel et al. and Voita et al.

Concerns

- Training routine seems very time consuming as the network is fine-tuned after removing every single attention head. The authors can try to remove the heads in a batch in order to reduce the training time.
- For reducing over-fitting simple methods like increasing the dropout probability or reducing number of transformer blocks could also be tried. It would be good if the authors can provide comprehensive comparison with simple techniques in order to justify the multi-step training for AUBER.

---

> ### Author Response · Authors · 2020-11-24
> **Thank you for your kind review on our paper.**
>
> Thank you for your kind and thorough review on our paper.
>
> 1.	Training routine seems very time consuming as the network is fine-tuned after removing every single attention head. The authors can try to remove the heads in a batch in order to reduce the training time.
> - As pointed out, the time consumption was greater than the other competitors and your solution would be effective for our time efficiency issue. However, due to the lack of time, we could not conduct the experiments and compare the method with the original one. Nevertheless, we appreciate for your brilliant suggestion.
>
> 2.	For reducing over-fitting simple methods like increasing the dropout probability or reducing number of transformer blocks could also be tried. It would be good if the authors can provide comprehensive comparison with simple techniques in order to justify the multi-step training for AUBER.
> - We compared our method only with the attention head pruning methods since attention head pruning algorithms can be used along with other regularization methods.

---

### Official Review · AnonReviewer1 · 2020-10-28
**Interesting ideas but weak experiments, requiring more discussions with related works**

**Rating:** 4
**Confidence:** 5

**Review:**

This paper proposes AUBER which applies reinforcement learning algorithms (DQN) to progressively prune attention heads from the lower layer to the higher layer in pre-trained transformer models (BERT) in order to improve the model fine-tuning. The state of DQN is the L1 norm of the value matrix of each attention head, the action space for each layer is the total number of attention heads with an additional quit action (H + 1).  To reduce the search space, AUBER prunes one head each time until it reaches the quit action in the lower layer and then repeats the process in the next layer. Experiments on 4 tasks from GLUE shows the effectiveness of AUBER against baselines. Overall, it is quite interesting and inspiring.

Here are a few concerns.
--There are many works such as [1, 2, 3, 4, 5, 6] and adapter-based approaches to improve the BERT fine-tuning. It is better to have a discussion compared with those algorithms.

--The 4 tasks evaluated are small. What’s the performance on the larger tasks, such as MNLI? Does AUBER also help in these settings? I’d like to mention that there are some issues with WNLI and all the machine learning algorithms without specific additional data or hack cannot outperform a simple majority voting. Thus it is better to select other tasks for evaluation. Lastly, these tasks have large variance and I’d like to suggest reporting mean/var instead of a single number.

--AUBER is complicated and requires L (# of layers) rounds to get the final model. Thus, it is better to report the comparison of training time between these baseline systems. Does AUBER use the dev set to compute rewards during training? If so, it is not fair to report the dev accuracy on these models.

--Pruning attention heads in BERT is interesting. However, the attention block only takes  around 25% of total parameters of BERT models. Is there any way to apply the current approach to other components in BERT?

--In section 4.1, it claims that fine-tuning on small tasks e.g., RTE often fails. It is better to provide more evidences, since a simple fine-tuning approach on BERT leads SOTA on these tasks compared with the pre-BERT era models/algorithms.



[1] Zhang et al. Revising Few-sample BERT fine-turning, https://arxiv.org/abs/2006.05987

[2] Zhu et al. FreeLB: Enhanced adversarial training for natural language understanding, ICRL 2020.

[3] Jiang et al. Smart: Robust and efficient fine-tuning for pre-trained natural language models through principled regularized optimization, ACL 2020.

[4] Liu et al. Multi-task deep neural networks for natural language understanding, ACL 2019.

[5] Prasanna et al. When BERT Plays the Lottery, All Tickets Are Winning, https://arxiv.org/abs/2005.00561

[6] Chen et al. The Lottery Ticket Hypothesis for Pre-trained BERT Networks, https://arxiv.org/abs/2007.12223

---

> ### Author Response · Authors · 2020-11-24
> **Thank you for your concern on our work**
>
> Thank you for your time and concern on our work. We carefully reviewed your feedbacks and referred a lot of it to revise our paper.
>
> 1.	There are many works such as [1,2,3,4,5,6] and adapter-based approaches to improve the BERT fine-tuning. It is better to have a discussion compared with those algorithms.
> - In this paper, we limited the competitors to the attention head pruning methods as our algorithm can be used along with other regularization methods such as dropout. Moreover, we think that attention head pruning has its advantage in high explainability. Thus, we believe that comparing our method to general parameter pruning algorithms can be a bit unfair comparison.
>
> 2.	The 4 tasks evaluated are small. What’s the performance on the larger tasks, such as MNLI? Does AUBER also help in these settings? I’d like to mention that there are some issues with WNLI and all the machine learning algorithms without specific additional data or hack cannot outperform a simple majority voting. Thus it is better to select other tasks for evaluation. Lastly, these tasks have large variance and I’d like to suggest reporting mean/var instead of a single number.
> - We repeated the experiments 5 times and added the results to the paper. Please refer to our updated version of the paper. Also, we are sorry to say that we could not conduct the experiments with other GLUE datasets because of the lack of time.
>
> 3.	AUBER is complicated and requires L (# of layers) rounds to get the final model. Thus, it is better to report the comparison of training time between these baseline systems. Does AUBER use the dev set to compute rewards during training? If so, it is not fair to report the dev accuracy on these models.
> - Just like your concern, AUBER takes much longer training time compared to the other competitors. However, since our focus is on the regularization performance not the time efficiency of the algorithm, we excluded the experiments to measure the training time. Moreover, we do not use the dev set at all for reward computation. Instead, we split the training dataset into two sets: one for the fine-tuning while the other for the reward computation. By splitting the training set into two, we are able to compute the reward without exploiting any of the dev dataset.
>
> 4.	Pruning attention heads in BERT is interesting. However, the attention block only takes around 25% of total parameters of BERT models. Is there any way to apply the current approach to other components in BERT?
> - Since our algorithm is devised based on the BERT attention heads’ unique structure, extensive modification will be required to utilize the method to the other components. In this paper, we focused only on the attention head pruning since its head-wise and layer-wise structures yield relatively simple RL model and better explainability. However, we believe that reinforcement learning can be used for pruning or regularization of the other components in BERT too just as it is used in CNN.
>
> 5.	In section 4.1, it claims that fine-tuning on small tasks e.g., RTE often fails. It is better to provide more evidences, since a simple fine-tuning approach on BERT leads SOTA on these tasks compared with the pre-BERT era models/algorithms.
> - As the reference [1] has already stated the issue, we did not address it in the paper.
>
> [1] Phang et al., Sentence encoders on STILTS: Supplementary training on intermediate labeled-data tasks, CoRR, abs/1811.01088, 2018

---

### Official Review · AnonReviewer4 · 2020-10-29

**Rating:** 4
**Confidence:** 4

**Review:**

========================

Paper Summary:

This paper proposed to prune heads of multi-head attention in BERT to achieve regularization for tasks with a smaller dataset. Existing pruning papers on transformers focus on compression (model size) with a smaller loss in accuracy but this work claims that pruning can actually improve accuracy in some cases. The authors use DQN to learn a policy to prune attention head layer by layer. They demonstrate improvements on 4 small tasks from GLUE.

========================

Review

The central idea of this paper is interesting, but the experiment is not convincing enough. Whether we actually need to prune large pretrained transformer for regularization itself is debatable because researchers have found that over-parameterized deep neural language models, for example GPT-3, is good at few shot learning. This paper needs much stronger experiments to support their claim.

Pros:
- The proposed method is interesting. Use pruning to improve performance is a new thing, at least with transformers and BERT.

Cons:
- This work is limited to small tasks. No results are demonstrated on larger datasets.
- Experiment is weak. The authors may improve this by running all tasks in GLUE, perhaps with a smaller training split to fit the authors’ setting. Also, it might make more sense to experiment on BERT-large or other larger, more powerful pretrained transformers to demonstrate the proposed idea can regularize over-parameterized networks.
- Lack of comparison with other regularization techniques, for example those mentioned in related work.
- The training overhead of this method is not discussed. It involves repeated finetuning after each layer is pruned. It will benefit the readers to measure the time needed for training AUBER.

==========================

Other Questions

- Is Table 2 a fair comparison with other baselines? In AUBER, model is finetuned after each layer is pruned. It is not clear if this finetuning is done for other heuristic-based pruning. This might need another ablation experiment.
- Why not directly use the whole value embedding V as the input to the DQN? Theorem 1 is also somewhat heuristic.
- It is not clear how the training data is sampled in Algorithm 1 to train the DQN model. Does s contain all training examples? Also, in line 29, what data does the model see to decide pruning?

=================================

Minor Issues

- Please fix the citation format (citep/citet in latex).
- Perhaps Figure 1 can be improved. In my opinion, the current figure does not help the readers understand the method.

---

> ### Author Response · Authors · 2020-11-24
> **We appreciate your comprehensive reviews**
>
> We appreciate your comprehensive reviews and advices on our paper. We want to note that we have conducted additional experiments according to your review and attached the results to the updated paper.
>
> 1.	This work is limited to small tasks. No results are demonstrated on larger datasets. Experiment is weak. The authors may improve this by running all tasks in GLUE, perhaps with a smaller training split to fit the authors’ setting. Also, it might make more sense to experiment on BERT-large or other larger, more powerful pretrained transformers to demonstrate the proposed idea can regularize over-parametrized networks.
> - Because of the resource constraints, we could not address larger models such as BERT-large. However, we appreciate your recommendation to conduct experiments with other models and GLUE tasks and will continuously attempt to do the experiments in order to bolster our method’s superiority.
>
> 2.	Lack of comparison with other regularization techniques, for example those mentioned in related work.
> - We limited the competitors to the ones which used pruning techniques for regularization.
>
> 3.	The training overhead of this method is not discussed. It involves repeated finetuning after each layer is pruned. It will benefit the readers to measure the time needed for training AUBER.
> - As you have stated, it takes longer time compared to the other methods. Nevertheless, since our main goal is to improve the performance not to reduce the training time, we focused on the performance enhancement and did not investigate the training time.
>
> 4.	Is Table 2 a fair comparison with other baselines? In AUBER, model finetuned after each layer is pruned. It is not clear if this fine-tuning is done for other heuristic-based pruning. This might need another ablation experiment.
> - Thanks to your review, we modified the experiment setting to fine-tune the model after each layer is pruned with the other baselines. The results are attached to the updated paper.
>
> 5.	Why not directly use the whole value embedding V as the input to the DQN? Theorem 1 is also somewhat heuristic.
> - We tried to minimize the size of input to DQN (state vector) so as to reduce the model complexity of the DQN.
>
> 6.	It is not clear how the training data is sampled in Algorithm 1 to train the DQN model. Does s contain all training examples? Also, in line 29, what does the model see to decide pruning?
> - We split the training dataset into two sets: a set for fine-tuning and the other for reward evaluation. We added the statement about the dataset split in the paper. When the model decides the optimal pruning policy in line 29 of Algorithm 1, it only takes the initial state vector as input and outputs the optimal pruning policy for the layer.
>
> 7.	Please fix the citation format.
> - Thanks to your advice, we reformatted the citation more concisely.
>
> 8.	Perhaps Figure 1 can be improved. In my opinion, the current figure does not help the readers understand the method.
> - We added a figure to illustrate how the algorithm prunes attention heads in a layer. It contains overall flow of our algorithm.

---

### Official Review · AnonReviewer3 · 2020-10-30
**Interesting work on BERT pruning using reinforcement learning**

**Rating:** 5
**Confidence:** 4

**Review:**

Summary: This work proposes learning to prune attention heads in BERT in order to achieve better accuracies on downstream tasks, especially when there are a small number of training examples. The authors employ reinforcement learning, or more specifically deep Q-learning, to learn the pruning policy in a layer-wise manner. The model is finetuned iteratively once pruning is done for each layer. Experiments show that after pruning through the proposed method, the performance improves. Ablations shows the designs of the state representation and order of pruning are effective.

Pros:
The paper is clearly presented and easy to follow. The use of RL for pruning policy is reasonable, and to my knowledge this is the first work on BERT models. The experiments shows the pruned models can achieve better performance than the original models on several benchmark datasets.

Concerns and questions:
1. The idea of using reinforcement learning for neural network pruning has been tackled in previous works and is thus not quite new.
2. The reason of choosing L1-norm of the value matrix as the state vector does not fully convinces me. The comparison in Table 3 may not be fair since different numbers of heads are pruned under different strategies, therefore the models have different capacities. A related question is that since the value matrix varies across training examples, will the state be the average of L1-norm of value matrix across all the training data? Please the author clarify.
3. Only four of the GLUE tasks are included in the experiments, it will be more interesting to know how the method works for other GLUE tasks with relatively large training data.
4. The performance of finetuning GLUE tasks could be unstable. Are the results reported in the paper from a single run or average of multiple runs? Is it possible the baseline model also have chance to achieve better performance with multiple runs?
5. AUBER finetunes models when heads are pruned in every layer, could this be the main reason that it works better than baselines? What if we conduct layer-wise finetuning for the baseline models?

---

> ### Author Response · Authors · 2020-11-24
> **Thank you for your detailed reviews.**
>
> Thank you for your detailed reviews. We have attached additional experimental results and explanations in the paper following your comments.
>
> 1.	The idea of using reinforcement learning for neural network pruning has been tackled in previous works and is thus not quite new.
> - As you have pointed out, this is yet the first attempt to use RL to prune and regularize transformers and BERT. Due to the unique structure that BERT has, we believe that pruning BERT model is very different from pruning general neural networks even though both of them exploit RL.
>
> 2.	The reason of choosing L1 norm of the value matrix as the state vector does not fully convince me. The comparison in Table 3 may not be fair since different numbers of heads are pruned under different strategies, therefore the models have different capacities. A related question is that since value matrix varies across training examples, will the state be the average of L1-norm of value matrix across all the training data? Please the author clarify.
> - We demonstrated that L1 norm of a value matrix is a great indicator of the role of the attention head in its layer. In the experiment summarized in Table 3, even though the model capacity is different due to the different number of heads pruned, the capacity is determined by the strategies. If a strategy is optimal, the strategy will make a good decision regarding the number of heads pruned. Thus, we think that the difference in the number of heads pruned does not deteriorate the fairness of the experiment. Moreover, we would like to clarify that the L1-norm of value matrix is the same across all the training data. As the value matrices are a part of the BERT model parameters, it does not differ depending on the input training data.
>
> 3.	Only four of the GLUE tasks are included in the experiments, it will be more interesting to know how the method works for other GLUE tasks with relatively large training data.
> - We have trained only with the datasets whose training data are less than 10,000. Also, resource constraints also blocked us from doing additional experiments. However, we appreciate your suggestion to examine the method with more tasks and will try our best to do additional experiments.
>
> 4.	The performance of fine-tuning GLUE tasks could be unstable. Are the results reported in the paper from a single run or average of multiple runs? Is it possible the baseline model also have chance to achieve better performance with multiple runs?
> - In order to show the stability of our method, we repeated the experiments 5 times for each dataset and reported mean and standard deviation of the results. It is attached in the table 2 of our newly updated paper.
>
> 5.	AUBER finetunes models when heads are pruned in every layer, could this be the main reason that it works better than baselines? What if we conduct layer-wise fine-tuning for the baseline models?
> - We changed the experimental setting to fine-tune the model in layer-wise manner with baseline models. Even though we fine-tune the competitor models every layer, our model still shows its superiority among the methods.

---

### Decision · Program_Chairs · 2021-01-07
**Final Decision**

**Decision:**

Reject

**Comment:**

This paper proposes to use RL to learn how to prune attention heads in BERT to achieve regularization for tasks with small dataset size. Specifically, the authors use DQN to learn a policy to prune heads layer by layer.

This paper receives 4 reject recommendations with an average score of 4.5. Though the idea in this paper is interesting, the experiments in the current draft are far from convincing. The reviewers have raised many concerns regarding the paper. (i) Experiments are weak. Only 4 GLUE tasks are considered; it is necessary to also test the proposed methods on other GLUE tasks. (ii) The comparison with other regularization techniques is lacking. (iii) The training overhead of this method needs more careful discussion, as it involves repeated finetuning after each layer is pruned, therefore could be very time-consuming. (iv) More comprehensive related work discussion is needed.

The rebuttal unfortunately did not address the reviewers' main concerns. Therefore, the AC regrets that the paper cannot be recommended for acceptance at this time. The authors are encouraged to consider the reviewers' comments when revising the paper for submission elsewhere.